# 19-(Benzyloxy)-19-oxojolkinolide B (19-BJB), an *ent*-abietane diterpene diepoxide, inhibits the growth of bladder cancer T24 cells through DNA damage

Ke Wang[1,2,3©], Juan-Cheng Yang[4©], Yeong-Jiunn Jang[4], Guan-Yu Chen[4], Ya-Jing Zhang[1,2], Yun-Hao Dai[4], Da-Yong Zhang[1,2]*, Yang-Chang Wu[3,4,5]*

1 Department of Medicinal Chemistry, China Pharmaceutical University, Nanjing, PR China, 2 Center for Drug Discovery, China Pharmaceutical University, Nanjing, PR China, 3 Graduate Institute of Integrated Medicine, China Medical University, Taichung, Taiwan, 4 Chinese Medicine Research and Development Center, China Medical University Hospital, Taichung, Taiwan, 5 Department of Medical Laboratory Science and Biotechnology, College of Medical and Health Science, Asia University, Taichung, Taiwan

© These authors contributed equally to this work.
* yachwu@gmail.com (YCW); zhangdayong@cpu.edu.cn (DYZ)

**Data Availability Statement:** All relevant data are within the manuscript and its Supporting Information files.

## Abstract

Diterpenoids jolkinolide A and B, were first isolated from *Euphorbia fischeriana*. In our previous research, 19-(Benzyloxy)-19-oxojolkinolide B (19-BJB), a derivative of jolkinolides, was synthesized as a novel *ent*-abietane diterpene diepoxide. In this study, 19-BJB showed strong *in vitro* activity against bladder cancer cell lines. DNA damage which was observed through the interaction of 19-BJB with nucleotide chains and affected DNA repair resulted in the activation of checkpoint kinase 1 (Chk1) and checkpoint kinase 2 (Chk2) in bladder cancer cell lines. *In vivo* testing in nude mice also proved that 19-BJB revealed a potential inhibitory effect on tumor growth. Additionally, the 3D-QSAR models of jolkinolides were established. Briefly, we proved that 19-BJB could potentially be used as a drug to inhibit the growth of bladder tumor.

## Introduction

Urothelial carcinoma is a major global health problem. Bladder cancer, a subtype of urothelial carcinoma, is among the top ten most common cancer types in the world. Each year, about 3.0% of the newly cancer diagnoses and 2.1% of cancer deaths are caused by urinary bladder cancer [1, 2]. The increasing number of bladder cancer patients in recent years due in part to the rise in the levels of environmental carcinogens [3]. It is estimated that the annual incidence of urothelial carcinoma in western countries is approximately two new cases per 100,000 inhabitants [4]. The therapeutic regimen for urothelial carcinoma relies on transurethral resection in combination with gemcitabine for local treatment [5]. Nevertheless, cisplatin is the standard chemotherapeutic agent for the treatment of advanced bladder cancer. However,

**Funding:** This research was funded by the National Natural Science Foundation of China (No. 30973607; No. 81172934); Ministry of Science and Technology of Taiwan (MOST 103-2911-I-002-303; MOST 104-2911-I-002-302; MOST 107-2320-B-037-001), National Health Research Institutes of Taiwan (NHRI-EX104-10241BI).

**Competing interests:** The authors declare no conflict of interest. The funders had no role in the design of the study; in the collection, analyses, or interpretation of data; in the writing of the manuscript, or in the decision to publish the results.

renal toxicity and chemoresistance can compromise the efficacy of cisplatin, indicating the need to find new agents that can improve the outcomes for patients with a poor prognosis.

Natural products have played a major role in new drug discovery for centuries, with over 47% of approved anticancer agents being of natural origin [6]. Due to the special structures and diverse biological properties, diterpenoids have attracted much attention of researchers [7]. Plants belonging to *Euphorbiaceae* families are widely applied in traditional Chinese medicine because these families have rich sources of diterpenoids [8]. Interestingly, the most recent reports have highlighted their potential as potent remedies against several cancers [9]. The abietane diterpenes, many of which have a α,β-unsaturated lactone functional group, are the main reason why *Euphorbiaceae* families have antitumor activities [10]. Jolkinolides A and B, a kind of diterpenoid, were first separated from *Euphorbia fischeriana* [11]. Jolkinolide B has been found to exhibit significant inhibitory effects, such as the activities against human prostate adenocarcinoma LNCaP cells [12], human leukemia K562 cells [13], human esophageal carcinoma Eca-109 cells, and human hepatoma carcinoma HepG2 cells [14]. Several anticancer mechanisms of jolkinolide B have been proposed. In human leukemic U937 cells, jolkinolide B was found to reduce cell viability and induce apoptosis in a dose-and time-dependent manner through the activation of caspase-3 and caspase-9, and the down-regulation of PI3K/Akt [15]. Ma et al. found that jolkinolide B inhibits RANKL-induced osteoclastogenesis by suppressing the activation of NF-κB and MAPK signaling pathways [16]. Later, Yang et al. found that jolkinolide B markedly attenuates LPS-induced histological alterations, lung edema, inflammatory cell infiltration, and myeloperoxidase activity as well as the production of TNF-α, IL-6, and IL-1β. Furthermore, jolkinolide B also significantly inhibits the LPS-induced the degradation of IκBα and phosphorylation of NF-κB p65 and MAPK [17]. The anticancer mechanisms of some jolkinolide B derivatives have also been studied. It was reported that 17-acetoxyjolkinolide B inhibits cytokine-induced NF-κB signal transduction [18]. Another derivative, 17-hydroxyjolkinolide B, was found to capable of inhibiting JAK/STAT3 signal transduction in HepG2 cells [19], while was found to exhibit LPS-induced production of inflammatory mediators such as prostaglandin E2, nitric oxide, and pro-inflammatory cytokines through the suppression of MAPK phosphorylation and NF-κB activation [20].

Recently, our research group reported a protocol for synthesizing jolkinolide A and jolkinolide B [21]. We also reported the first synthesis of jolkinolide derivatives [22, 23]. 19-(Benzyloxy)-19-oxojolkinolide B (19-BJB), one of derivatives we synthesized, showed potential inhibitory effect in different kinds of cancer cell lines. In this study, we have elucidated that 19-BJB showed strong *in vitro* activity against bladder cancer cell lines. This compound inhibited cell proliferation and induced DNA damage. The *in vivo* testing in nude mice also proved that 19-BJB had a potential inhibitory effect on tumor growth. Additionally, the 3D-QSAR models of jolkinolides were established and the bulky groups on C-19 of ring A maybe the key moiet of DNA damage. All of the compounds described above are shown in Fig 1 [11, 18–23].

## Materials and methods

The 5-week BALB/c male nude mice were from National Laboratory Animal Center (Taipei) and were fed with standard procedures in the SPF area of the Animal Experimental Center of China Medical University. Approval number: 2016–238. See in NC3Rs ARRIVE Guidelines Checklist and Original ARRIVE Guidelines (Affidavit of Approval of Animal Use Protocol).

### Cell lines, compounds, and reagents

T24, J82, NTUB1, NP14 and NTaxol cell lines were obtained from National Taiwan University. T24 is a muscle-invasive, epithelial-like, adherent cell line derived from transitional cell

**Fig 1. Chemical structures of jolkinolides.**

carcinoma of urinary bladder tissue. J82 is a non-muscle-invasive, fibroblast-like, adherent bladder cancer cell line. NTUB1 is a muscle-invasive, epithelial-like, adherent cell line derived from a 70-year-old female patient. The cisplatin resistant type and the taxol resistant type are NP14 and NTaxol, which have the same cellular phenotypes as NTUB1 [24]. HaCat is an immortal keratinocyte cell line from adult human skin. All cells were cultured at 37˚C in a humidified incubator with 5% $CO_2$.

Tested compound 19-BJB, 19-(benzyloxy)-19-oxojolkinolide B, was synthesized by our lab with its purity > 98%, dissolved in dimethyl sulfoxide (DMSO), aliquoted and stored at—20˚C. KU-55933 (S5 Fig), the ATM (ataxia telangiectasia mutated) inhibitor (ATMI), was purchased from Sigma-Aldrich and dissolved in DMSO. Hydrochloric acid (HCl), DMSO, ethanol, ethylenediaminetetracetic acid disodium ($Na_2$EDTA) and tetrasodium ($Na_4$EDTA) salts, sodium chloride (NaCl), and sodium hydroxide (NaOH) were purchased from Carlo Erba Reagenti Srl. 0.5% trypsin-EDTA, FBS, penicillin-streptomycin were purchased from Gibco. Caspase-3 rabbit mAb, Chk1 mouse mAb, Chk2 rabbit mAb, cleaved caspase-3 rabbit mAb, Cleaved PARP-1 antibody, PARP-1 antibody, phospho-Chk1 rabbit mAb, phospho-Chk2 antibody, phospho-histone H2A.X rabbit mAb, propidine iodide (PI), protease inhibitor were purchased from Cell Signaling. 3-(4,5-dimethyl-2-thiazolyl)-2,5-diphenyl-2H-tetrazolium bromide (MTT), DMEM/F12, doxorubicin, phenylmethanesulfonyl, PMSF, RPMI-1640, sodium bicarbonate, sodium orthovanadate ($Na_3VO_4$) were purchased from Sigma. Goat anti-mouse IgG-HRP, goat anti-rabbit IgG-HRP, β-actin antibody were purchased from Santa Cruz. Sodium fluoride, NaF, Triton X-100 and RIPA lysis buffer were purchased from Merck. Ammonium persulfate (APS) and TEMED were purchased from J.T.Baker. PBS, sodium

dodecyl sulfate (SDS), tris (hydroxymethyl) aminomethane (Tris base) and skimmed milk powder were purchased from Hyclone, Millipore, GeneMark and Anchor, respectively.

## MTT assay

The cell lines were seeded in an atmosphere of 5% $CO_2$ at 37˚C. Cells were plated into 96-well plates at a density of $5\times10^3$ cells in 100 μL of growth medium 24 h prior to treatment. Following treatment with 19-BJB at different dosages (1.56, 3.13, 6.25, 12.50, 25, 50, and 100 μM) for 48 h, 100 μL of MTT solution (0.5 mg/mL) was added to each well and the plates were incubated at 37˚C for 1 h. Then the MTT solution was replaced by DMSO (100 μL) to dissolve the reduced MTT crystals. Cell viability was assessed by measuring absorbance at 550 nm in an ELISA Reader [25]. Dose response curves were then created as a percentage of vehicle treated control cells and $IC_{50}$ was calculated using GraphPad Prism 5. The procedures of the MTT tests for other drugs, including jolkinolide A, jolkinolide B, cisplatin, and paclitaxel, were the same as those for 19-BJB.

## Colony assay

The cells were maintained in 6-well plates at a concentration of 100 cells per well. Each well was treated with different concentrations of 19-BJB. The cultures were maintained under standard culture conditions. After two weeks, the medium was replaced with crystal violet solution and was stained for 5 min. The number of colonies was determined by an inverted phase-contrast microscope at ×40 magnification. A group of > 10 cells was considered as a colony. The colonies were counted 3 times, and the average was calculated.

## Live and dead

T24 cells were incubated in 6-well plate at $1 \times 10^5$ cells / well. 19-BJB was added at the concentrations of 0 μM, 0.5 μM, 1 μM, 2 μM, 4 μM, and 8 μM. After 48 h incubation, trypsin was added, and the cells were washed twice by PBS, then centrifuged at 1000 rpm for 5 min. 2 μM calcein AM and 4 μM ethidium homodimer were added in 1 mL PBS as dye to stain the cells, which were then observed using a TailTM Image-Based Cytometer 30 min later. In the further experiment, cells were treated with 10 μM ATMI for 3 h before 19-BJB (0, 4, and 8 μM) was added in different dosage. The cells were maintained in the medium with ATMI and 19-BJB for 48 h, and were harvested for the live and dead test.

## FACS analysis of cell cycle

Once T24 cells achieved 70% to 80% confluency, they were treated with 0.1% DMSO or 19-BJB at concentrations of 0 μM, 2 μM or 4 μM for 24 h and 48 h. After treatment, the cells were fixed in ice-cold 70% ethanol overnight. After fixation, the cells were washed three times with cold PBS and then stained in 500 μL of propidium iodide solution. Samples were analyzed on a BD FACScan flow cytometer, and the percentages of cells in the G0-G1, S, and G2-M phases of the cycle were determined using WinMDI 2.9.

## Protein isolation and western blot analysis

Samples (normalized according to cell number) were treated with 19-BJB at varying concentrations over 24 h or 48 h. Cell extracts were then prepared in 1000 μL RIPA lysis buffer containing protease inhibitors, 5 μL $Na_3NO_4$ (200 mM), 5 μL PMSF (200 mM) and 5 μL NaF (200 mM). Cell lysates were centrifuged at 4˚C and 15000 rpm for 30 min, and the supernatant was collected. BCA assay was used to test protein concentration in cells. Clarified protein lysates

containing equal amounts of protein (20 μg) were separated on 8–15% sodium dodecyl sulfate-polyacrylamide gel electrophoresis (SDS-PAGE) gels and electrophoretically (2 h at 90 V) transferred to a PVDF membrane. Blots were then blocked for 1 h in TBST containing 5% blocking grade non-fat dry milk, and then incubated overnight with primary antibody at 4˚C. Blots were then washed in TBST three times and incubated for 1 h at room temperature with secondary antibody. Electrochemiluminescence was used to enhance the detection of immunoreactive bands.

## Genotoxicity testing

The information about DNA damage given by the comet assay reflects the number of single or double stand breaks formed in the cellular DNA before or during the process of electrophoresis [26]. For this test, a suspension of isolated cells is embedded into an agarose gel onto a microscope slide and subsequently lysed by detergents in lysis buffer. Then the liberated DNA is exposed to alkali to unwind it from the strandbreakage sites and electrophoresed under alkaline conditions. In the presence of DNA strand breaks, staining with propidium iodide solution results in structures resembling comets with the tail length or tail fluorescence content reflecting the frequency of DNA strand breaks and hence DNA damage [27]. The standard alkaline procedure allows the detection of both single- and double-strand DNA breaks as well as apurinic/apirimidinic sites that are expressed as frank strand breaks in the DNA under the alkaline conditions of the assay. Immediately after exposure, cells were processed in the comet assay under alkaline conditions, basically following the original procedure [28] with minor modifications [29]. The cells were seeded in 6-well plate and treated with 19-BJB for 48 h. Then the cells were washed twice with PBS and detached with 300 μL of 0.05% trypsin solution. About 2 min later, trypsinization was terminated by adding culture medium (1 mL). The cells in each treatment well were taken and collected by centrifugation at 1000 rpm for 5 min. 10 μL of PBS with pellets suspended was added into 100 μL LMA at 37˚C, and 60 μL was immediately spread onto microscope slides. Agarose microgels were set for 30 min at 4˚C. Then the slides were immersed in ice-cold lysis solution (10 mM Tris-HCl, 2.5 M NaCl, 100 mM Na2EDTA, and 1% Triton X100; pH 10) and left to stand for 1 h at 4˚C, protected from light. The slides were then washed twice with dd water and placed in an electrophoresis tank after lysis. Electrophoresisi was then performed for 10 min at 20 V. Then the slides were washed twice with dd water and fixed in 75% ethanol for 5 min. The slides were then dried and stained with propidium iodide for 5 min. The samples were then observed using a fluorescence microscope. The extent of induced DNA damage was measured as the percent of fluorescence migrated to the comet tail by a computer-based image analysis system (CometScore 15).

## Spectral pattern of 19-BJB synthesized on DNA template

The interaction of 19-BJB with DNA was studied using UV-visible absorption spectroscopy [30]. The stock DNA (1.0 mg/mL) was dissolved in Tris- HCl buffer (50 mM, pH = 7.5). 19-BJB was prepared at different concentrations (0, 5, 10, and 40 μM). 19-BJB and DNA were mixed and incubated at 37˚C for 2 h and 4 h. Then the mixture were dropped into a 1-cm quartz cell and scanned in the wavelength range of 220–350 nm at 25˚C using a Thermo-Scientific spectrophotometer (model Evolution 300, USA).

## 8-Oxoguanine detection

Cells were cultured in 6-well plates at $1 \times 10^5$ cells / well. Each well was treated with 19-BJB at varying concentrations (2 μM, 4 μM, and 8 μM) for 24 h. Then the cells were washed twice

with PBS and fixed with ice-cold 70% ethanol overnight. After fixation, the cells were washed thrice with cold PBS and then centrifuged at 1000 rpm for 5 min. The pellets were collected and 1% Triton and RNase A (10 mg/mL) were added at room temperature for 1 h. Anti-oxo-guanine 8 fluorescent antibody was added and the suspension were then stored at 4°C overnight. Then the samples were centrifuged at 1800 rpm for 5 min. After washing the samples twice with PBS, goat anti-mouse IgG FITC was added for 1 h. After washing the samples again with PBS, the fluorescence of the samples was analyzed using BD FACScan flow cytometer.

## The molecular docking study

The molecular docking calculation was performed by Autodock version 4.2 with the Lamarck-ian Genetic Algorithm to estimate the binding ability and simulate the binding model between 19-BJB and DNA (CGATCG) [31]. The structure of the DNA fragment (PDB ID: 1Z3F) was obtained from the Protein Data Bank (https://www.rcsb.org/) [32]. In the DNA structure, the co-crystalized substrates, including ligands, water and metal molecules were removed, and the polar hydrogens and the Kallman united atom charges were added to the atoms of the DNA by AutoDock Tool version 1.5.4 interfaces (ADT) [33]. The structural optimization of 19-BJB was performed by energy minimization with the MMFF94 force field using ChemBio3D software (version 11.0; Cambridge Soft Corp.) [34]. In addition, the polar hydrogens and Gasteiger charges were also added to the structure of 19-BJB by ADT [35]. The grid box calculated by the AutoGrid program was set at the location of the co-crystal ligand-binding site and its size was set bigger enough to contain the size of the ligand. Therefore, the coordinates of the central grid box were set at x = 0.395, y = 17.235, and z = 46.179, and the sizes of dimensions of x, y, and z-axis were respectively set as 56 x 38 x 38 Å grid at the spacing of 0.375Å [36]. All of the docking parameters were set to the default values except for the maximum number of energy evaluations, which was increased to 25,000,000 per run [37]. The docking results were analyzed with cluster analysis by ADT. Besides, all pictures were generated by the Accelrys Discovery Studio Visualizer (Version 4.0, Accelrys Software Inc.) [34].

## Toxicity test of 19-BJB

19-BJB was dissolved in DMSO and mixed with 0.9% physiological saline to the desired concentration. Two male nude mice were weighed and treated with 19-BJB at the dosage of 10 mg/kg three times per week. One mouse was treated by intraperitoneal injection while the other was treated by intravenous injection. After 4 days, the dosage was changed to 20 mg/kg. After 14 days, it was changed to 40 mg/kg. Both mice exhibited no signs of acute toxicity.

### *In vivo* antitumor efficacy study

Compound 19-BJB was tested for *in vivo* efficacy in 5-week BALB/c male nude mice xenograft models. T24 cells in the logarithmic growth phase were cultured and the injection volume was calculated at $1 \times 10^7$ cells / 0.2 mL / mouse. With the medium aspirated, the cells were washed with PBS. After being treated with trypsin, the cells were collected and centrifuged at 1000 rpm for 5 min, and then mixed with matrix gel with medium [38]. The mixed gel was injected into 14 mice averagely. The injected mice were randomized into control (saline) and treatment (19-BJB) groups (n = 7) when the volumes of xenografts were increased almost 50 mm$^3$. The mice were treated with normal saline (control group) or 19-BJB (20 mg/kg, treatment group) every two days. The tumor growth was then monitored daily beginning several days after the first injection. The tumor volume were measured as V (mm$^3$) = ab$^2$/2 (a = largest diameter of tumor, b = smallest diameter of tumor). All the mice were sacrificed with carbon dioxide after

28 days. The tumor tissues were washed in PBS after dissection, and were soaked in 10% para-formaldehyde for immunohistochemistry (IHC).

## Immunohistochemistry (IHC)

Tumors resected from mice were fixed in 10% paraformaldehyde for 48 hours and embedded in paraffin. 4-μm thick sections were cut and processed for immunohistochemistry and were incubated with anti-p-Chk1, anti-p-Chk2, anti-cleaved caspase-3 and anti-cleaved PARP-1, anti-TUNEL, and anti-Ki-67 (used at 1:100) at 4˚C overnight [39]. Following incubation, immunoperoxidase staining was carried out using a streptavidin-peroxidase kit obtained from Abcam. The slides were examined under a light microscope, and representative pictograms were taken from a minimum of five or six different slides for each group.

## CoMFA analyses of jolkinolide derivatives

To further understand the relationship between the structure and activity in jolkinolide deriva-tives, the comparative molecular field analysis (CoMFA) was performed by Sybyl-X version 1.2 software from Tripos Inc. (St. Louis, MO). Firstly, the structural optimization of 33 jolkino-lide derivatives was performed by energy minimization with MMFF94 force field using Sybyl-X software. Subsequently, 19-BJB was chosen as the template and then all the synthesized derivatives were aligned to the A and B rings of jolkinolides, set as core structure, by database alignment method using Sybyl-X software. To establish the CoMFA model, the anticancer activities of NTUB1 cell line of 33 jolkinolide derivatives were conducted and then transferred the $pIC_{50}$ (-log $IC_{50}$) for calculation. Steric and electrostatic field in three-dimensional grid with a spacing of 2.0 Å were calculated for all derivative compounds at each lattice intersection. A sp3 carbon atom was used as a probe atom with a charge of +1.00 charge, and a cutoff energy value of +30 kcal/mol was used to minimize electrostatic energies. A partial least-squares (PLS) method was used to linearly correlate the CoMFA fields to the values of $pIC_{50}$. In addi-tion, the leave-one-out (LOO) validation method and cross-validate (CV) analysis were per-formed to validate the quality of the CoMFA model and the regression of model was performed by no validation analysis.

## Statistical analysis

The data are presented as mean ± SD for triplicate samples. GraphPad Prism 6 (GraphPad Software Inc., San Diego, CA, USA) was used to analyze the significance of the differences among the groups. Unpaired two-tailed t test was performed to analyze the differences between two independent groups. * $p < 0.05$, ** $p < 0.01$, *** $p < 0.001$, **** $p < 0.0001$ were considered statistically significant.

# Results

## 19-BJB inhibits proliferation of human bladder cancer cells

To investigate the effects of 19-BJB on cell growth, T24, J82, NTUB1, NP14, Ntaxol and HaCat cells were exposed to eight different concentrations of 19-BJB for 48 h (Fig 2). Table 1 shows the $IC_{50}$ values of 19-BJB in the different cell lines. The results of MTT testing showed that 19-BJB had a better inhibition than the positive drug cisplatin, but a poorer inhibition than paclitaxel.

The colony formation assay showed that T24 cells formed significantly fewer colonies after treatment (Fig 3). Relatedly, it could clearly be seen that the treatment of T24 cells with 19-BJB resulted in a significant different speed of growth in a dose-dependent manner, with the

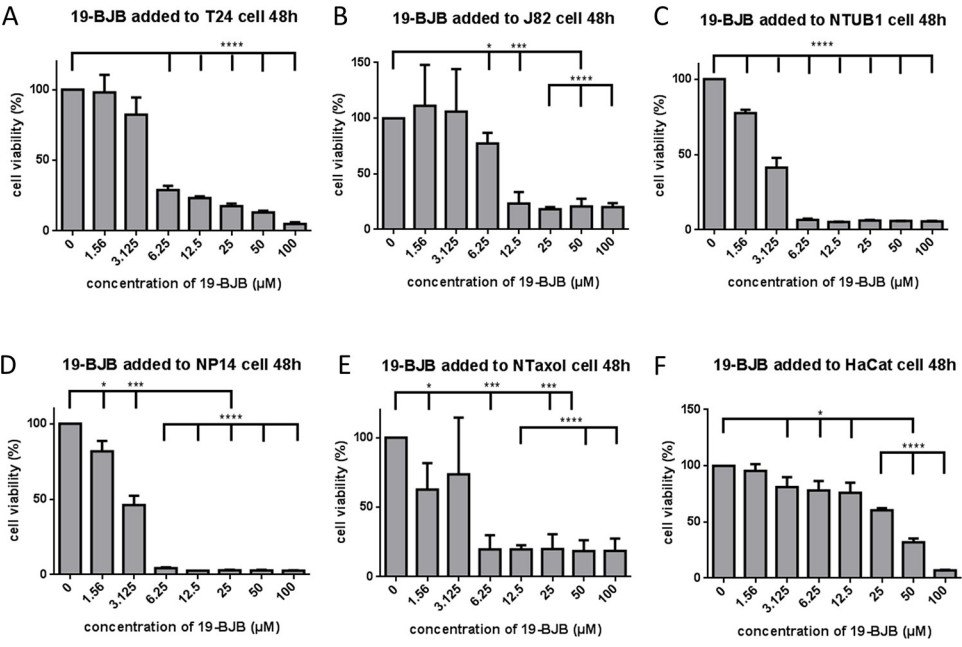

**Fig 2.** Effects of 19-BJB on cells viability in T24 (A), J82 (B), NTUB1 (C), NP14 (D), NTaxol (E), HaCat (F) cells. The cells were treated with different concentrations of 19-BJB for 48 h. The cell viabilities were measured by MTT assay as Table 1. Values are expressed as mean ± SD for triplicate samples and expressed as a percentage of control (set as 100%). * $p < 0.05$, ** $p < 0.01$, *** $p < 0.001$, **** $p < 0.0001$.

number of surviving colonies in the medium decreasing with higher concentration of 19-BJB. The cells could not survive in 19-BJB at the concentration of 8 μM.

We then used an image-based cytometer to analyze the percentage of live cells stained (Fig 4A and 4B) with calcein AM and dead cells stained with ethidium homodimer-1, respectively. The cell live and dead test showed that the results of cell staining were affected by 19-BJB in a dose-dependent manner. The results showed that the number of cells with green fluorescence were decreased and the number of cells with red fluorescence were obviously increased in the context of over 4 μM of 19-BJB (Fig 4C). Notably, the treatment of T24 cells with 19-BJB significantly induced cell death.

The results of MTT testing, colony assay, live and dead test indicated that 19-BJB had an inhibitory effect on the growth of T24 cells. In MTT test, the inhibitory effects of 19-BJB against cisplatin-resistant bladder cancer NP14 and paclitaxel-resistant bladder cancer NTaxol

**Table 1. *In vitro* inhibitory effect of jolkinolide A, jolkinolide B and 19-BJB in bladder cancer cells.**

| Compound | inhibitory effect after 48 h (IC$_{50}$ μM)[a] | | | | |
|---|---|---|---|---|---|
| | **T24[b]** | **J82[b]** | **NTUB1[b]** | **NP14[b]** | **NTaxol[b]** |
| jolkinolide A | 37.8±1.1 | 49.4±0.1 | 91.6±29.2 | not detected | 469.5±46.5 |
| jolkinolide B | 10.9±6.5 | 80.8±14.8 | 23.3±5.2 | 14.6±1.3 | 13.5±1.9 |
| 19-BJB | 6.0±1.3 | 12.6±6.0 | 2.2±0.2 | 2.3±0.3 | 4.6±3.8 |
| cisplatin | 4.2±0.3 | 1.9±0.3 | 5.4±0.4 | 69.0±22.9 | 6.1±1.0 |
| paclitaxel | 0.12±0.03 | 0.4±0.1 | 0.9±0.3 | 0.04±0.02 | 17.6±10.7 |

[a]Inhibition of cell growth by listed compounds was determined using MTT assay.

[b]T24, J82, and NTUB1 are human bladder carcinoma cell lines, NP14 is cisplatin resistant NTUB1 cell line, NTaxol is paclitaxel resistant NTUB1 cell line.

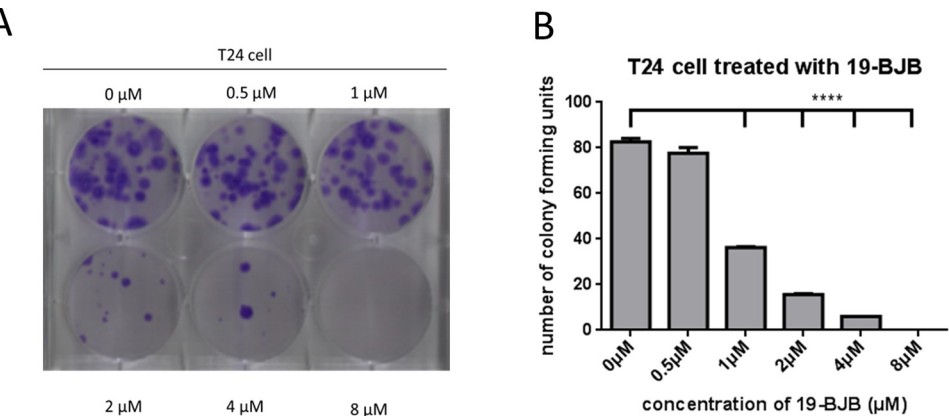

**Fig 3. Colony assay of T24 cells.** (A) The cells were analyzed by crystal violet solution stained for 5 min after 19-BJB treatment at varying concentrations for two weeks. The number of cells was significantly decreased by 19-BJB in a dose-dependent manner. (B) The data are expressed as means ± SD from triplicate samples for each treatment. ** $p <$ 0.01, *** $p < 0.001$, **** $p < 0.001$.

were much better than the effect of 19-BJB against its parental bladder cancer NTUB1. This finding prompt us the mechanism that 19-BJB induced in cancer cell growth may be totally different from that of cisplatin or paclitaxel. In previous reports, jolkinolide B was found to inhibit DNA synthesis, induce G1 phase arrest and lead to apoptosis in LNCaP cells [12].

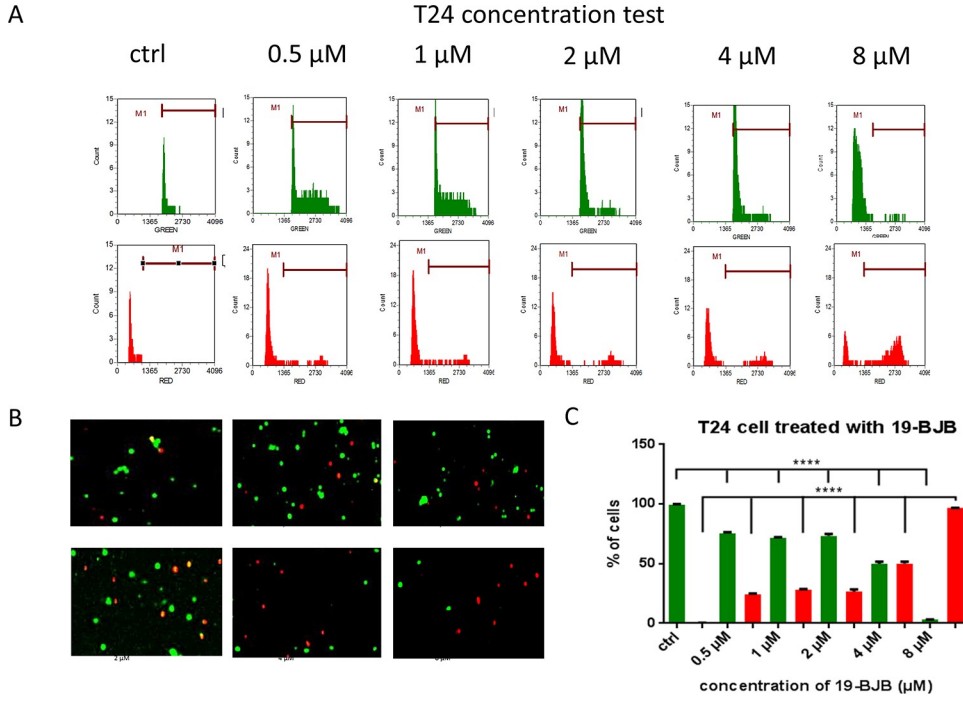

**Fig 4. The results of live and dead test of cells detected by TailTM Image-Based Cytometer.** Green column represented the number of live cells and red column represented the number of dead cells. (A) The raw data of fluorescence response of T24 cells after treated with different concentrations of 19-BJB for 48 h (0, 0.5, 1, 2, 4, 8 μM) was showed. (B) T24 live cells were stained green by 4 μM calcein AM and dead cells were stained red by 2 μM ethidium homodimer. The histogram data (C) of T24 are expressed as means ± SD from triplicate samples for each treatment. * $p < 0.05$, ** $p < 0.01$, *** $p < 0.001$, **** $p < 0.0001$.

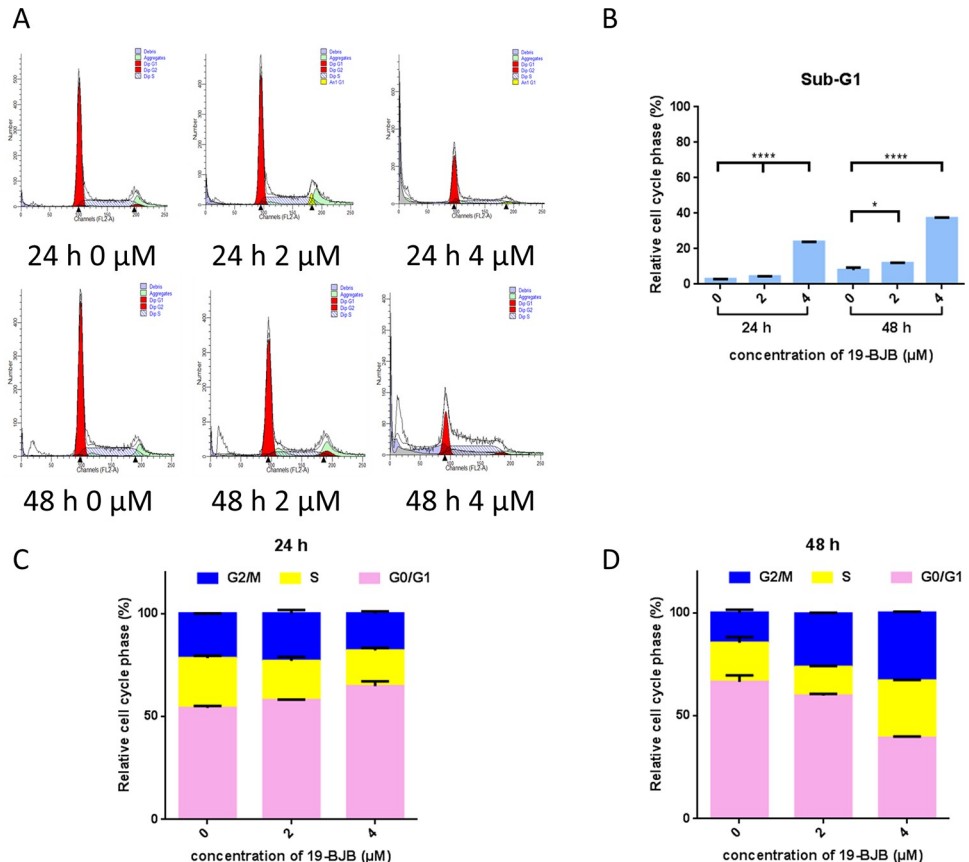

**Fig 5. 19-BJB induces cell cycle arrest in T24 cells.** (A) FACS analysis data of T24 cell cycle. Samples were analyzed on a BD FACScan flow cytometer and the percentage of cells in the Sub-G1, G0-G1, S, and G2-M phases of the cycle was determined using FACS Express V3. (B) The change of Sub-G1 phase in FACS analysis affected by 19-BJB. (C) T24 cells treated with 19-BJB (0, 2, 4 μM) for 24 h. (D) T24 cells treated with 19-BJB (0, 2, 4 μM) for 48 h. The data are expressed as means ± SD from triplicate samples for each treatment. *p < 0.05, ****p < 0.0001.

Jolkinolide B was also demonstrated to arrest the cell cycle in G1 phase and induce apoptosis in K562 cells [13]. As 19-BJB was a derivative of jolkinolide B, DNA damage study was considered in our research.

To examine whether 19-BJB treatment could affect cell cycle progression in bladder cancer cells, T24 cells were treated with different concentrations of 19-BJB. To further examine the effects of 19-BJB on cell cycle progression, an analysis of the collected cells by flow cytometry was performed (Fig 5A). The results for the cells treated for 24 h shown in Fig 5C indicated that the fractions of G0/G1 phase were increased as the treatment concentration of 19-BJB was increased. The fraction of G2/M in 2 μM group did not change obviously. While in 4 μM group, the fraction of G2/M decreased. In 48 h group (Fig 5D), the fractions of G0/G1 phase decreased. At the same time, the fractions of G2/M phase increased correlated with concentration. A large amount of cell debris formed as the treatment time was increased, which indicated that cell death may occur when the treatment time was 24 h. In 48 h group, the apoptosis peak became more obvious as the concentration of 19-BJB was increased (Fig 5B).

## DNA damage effect of 19-BJB

The comet assay was conducted to test the genotoxicity of 19-BJB. The information about DNA damage yielded by the comet assay indicates the number of single- or double- strand

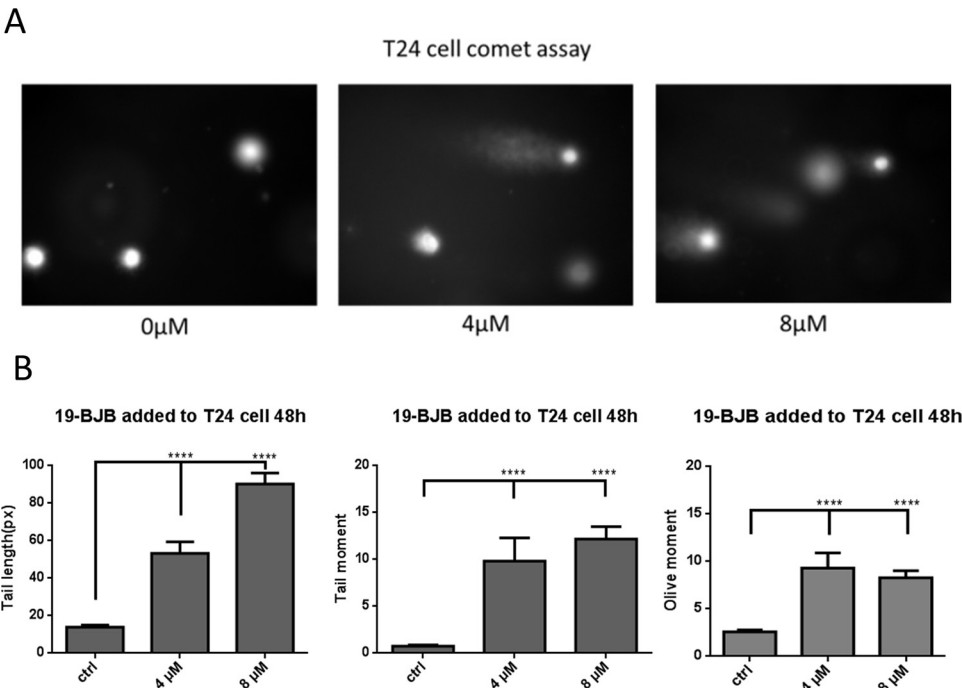

**Fig 6. Comet assay was performed T24 cells after 48 h of 19-BJB treatment.** (A) Photos observed using a fluorescence microscope. (B) Tail length, tail moment and olive moment of comet assay were measured and photomicrographs by CometScore 15. Each bar is representative of mean ± SEM of three replicates for each treatment (***p < 0.001).

breaks. DNA damage can be evaluated by morphological changes in the nucleus and the tail length after electrophoresis and propidium iodide staining. For 19-BJB, the three concentrations (0 μM, 4 μM, 8 μM) were tested for genotoxicity, and the results are shown in Fig 6. CometScore was used to calculate the tail length, tail moment, and olive moment. It was obvious that the tail length, tail moment, and olive moment were increased at high concentrations of 19-BJB, which proved that nuclear damage occurred in T24 cells.

The checkpoint effector kinases Chk1 and Chk2 played an important role in the DNA damage response. Fig 7 illustrates that 19-BJB treatment of T24 cells resulted in increased expression of phospho-Chk1 (p-Chk1) and phospho-Chk2 (p-Chk2), while down-regulating the expression of Chk1 and Chk2. Also, 19-BJB treatment increased, in a dose-dependent manner, the expression of cleaved caspase-3 and cleaved poly ADP-ribose polymerase (cleaved PARP-1), as well as the phosphorylation of H2A histone family member X (p-H2AX) in T24 cells, compared to vehicle-treated controls. Taken together, these results imply that 19-BJB activates the phosphorylation of DNA damage response-related proteins, such as Chk1, Chk2, and Histone H2AX, influencing the cleavage of caspase-3 and indicating apoptotic effects against bladder cancer cells.

We also examined the influence of ATMI to further elucidate the role of 19-BJB in DNA damage. Fig 8 showed the protein expression of p-Chk1, p-Chk2 and p-H2AX increased when 19-BJB (0, 2, 4 μM) exist, just similar to Fig 7. But when 10 μM ATMI was added before T24 cells was treated with 19-BJB, the expression of DNA damage related proteins were down-regulated. Taken together, these finding suggested that the increasing expression of p-Chk1, p-Chk2 and p-H2AX caused by 19-BJB could be reversed in the presence of ATMI.

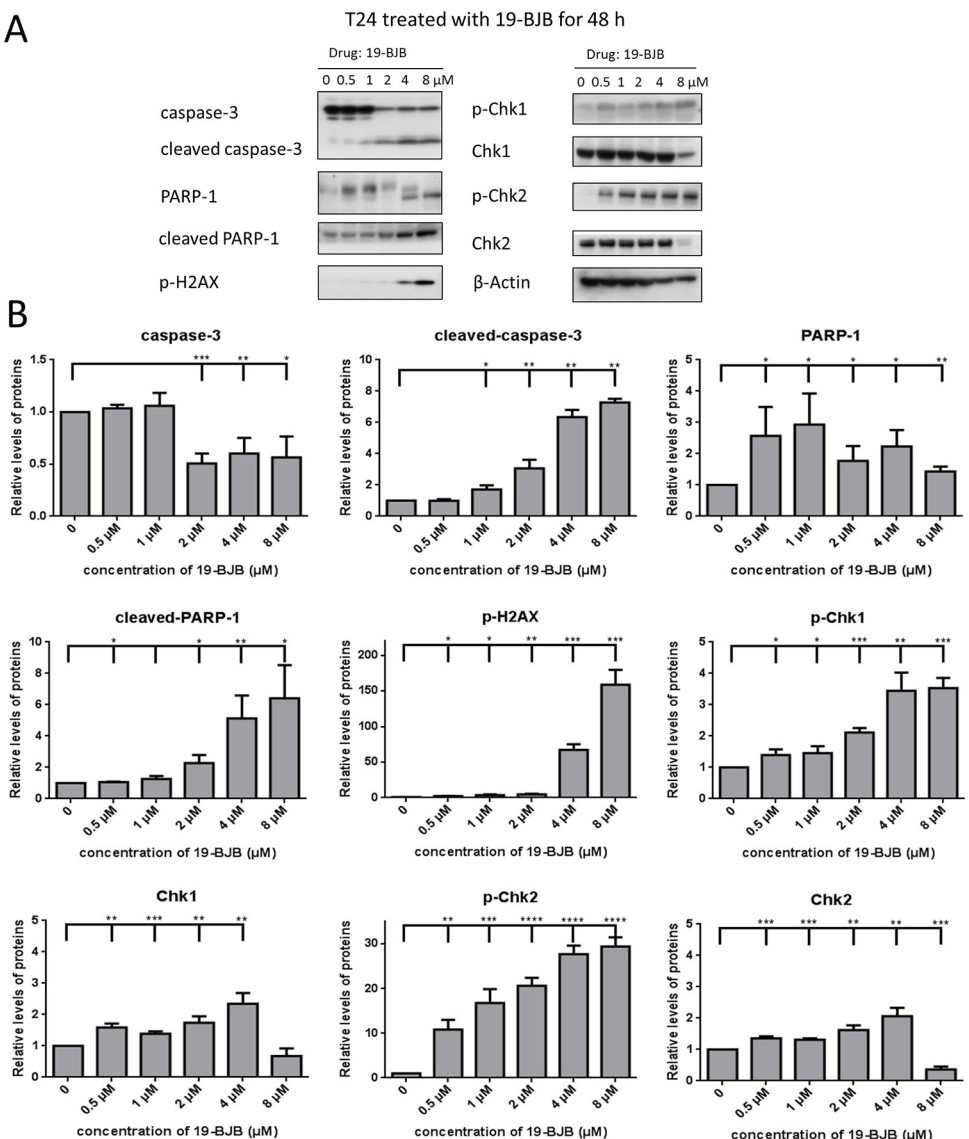

**Fig 7. Western blots analysis of 19-BJB.** T24 cells were treated with 19-BJB in varying concentration for 48 h and total cell lysates were analyzed for the expression of DNA damage repair related protein PARP-1, cleaved PARP-1, p-H2A.X, Chk1, Chk2, p-Chk1 and p-Chk2. The apoptosis related protein caspase-3 and cleaved caspase-3 were also analyzed (A). Equal protein loading was confirmed by reprobing membrane with β-actin after stripping. The relative levels of all proteins are performed (B) as means ± SD from triplicate samples for each treatment. $^*$p < 0.05, $^{**}$p < 0.01, $^{***}$p < 0.001, $^{****}$p < 0.0001.

In Fig 5, a small apoptosis peak was observed in concentration of 4 μM for 24 h group. Therefore, the western blot of DNA damage related proteins were also checked in Fig 9A. Cleaved-PARP-1 was increased in 4 μM and 8 μM groups. The expression of p-H2A.X was also significantly increased in 8 μM group.

Fig 8 suggested that ATMI would affect the expression of DNA damage response related proteins. To test the effect on cell survival, 10 μM ATMI was added 3 h before the cells treated with 19-BJB. The result of live and dead test in Fig 10 showed that ATMI increased the amount of dead cells both in 4 μM and 8 μM concentration group in T24 cell lines. The image-based cytometer data indicated that the amount of live cells was decreased when the medium was pretreated with ATMI.

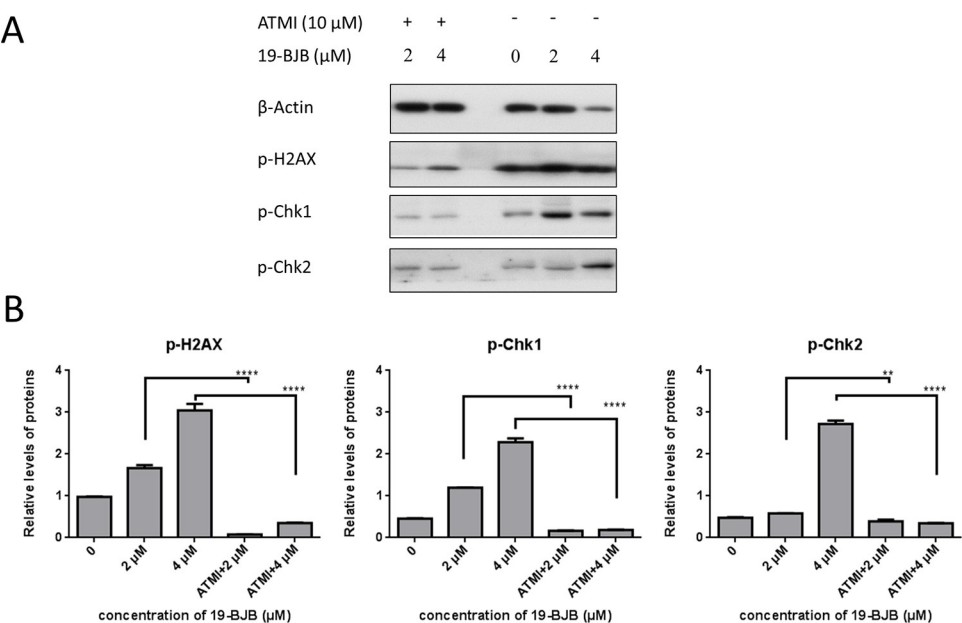

**Fig 8. Western blots analysis of 19-BJB combined with ATMI.** T24 cells were treated with 19-BJB in concentration of 0, 2, 4 μM for 48 h or ATMI (10 μM) 3 h before treating with 19-BJB for 48 h. And total cell lysates were analyzed for the expression of p-H2A.X, p-Chk1 and p-Chk2 (A). Equal protein loading was confirmed by reprobing membrane with β-actin after stripping. The relative levels of all proteins are performed (B) as means ± SD from triplicate samples for each treatment. **$p < 0.01$, ****$p < 0.0001$.

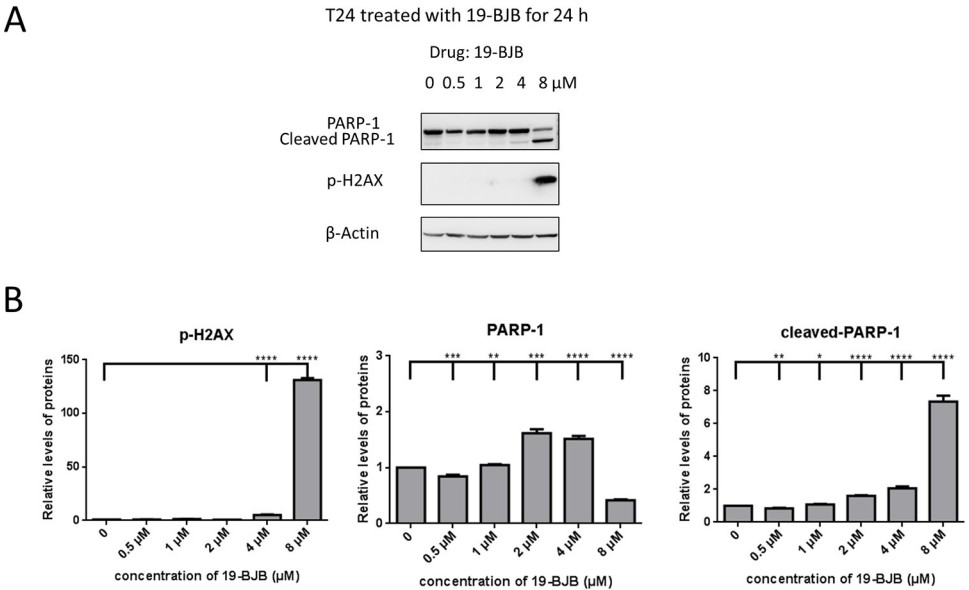

**Fig 9. Western blots analysis of 19-BJB in 24 h.** (A) T24 cells were treated with 19-BJB in varying concentration for 24 h and total cell lysates were analyzed for the expression of DNA damage repair related protein PARP-1, cleaved PARP-1, p-H2A.X. Equal protein loading was confirmed by reprobing membrane with β-actin after stripping. The relative levels of all proteins are performed (B) as means ± SD from triplicate samples for each treatment. *$p < 0.05$, **$p < 0.01$, ***$p < 0.001$, ****$p < 0.0001$.

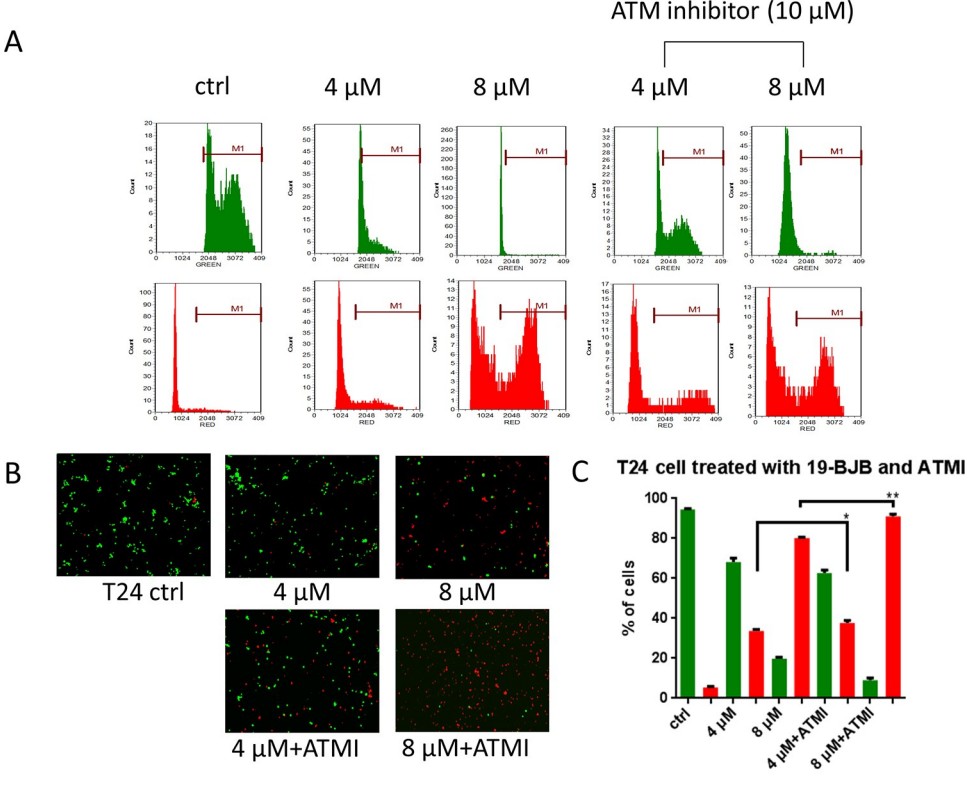

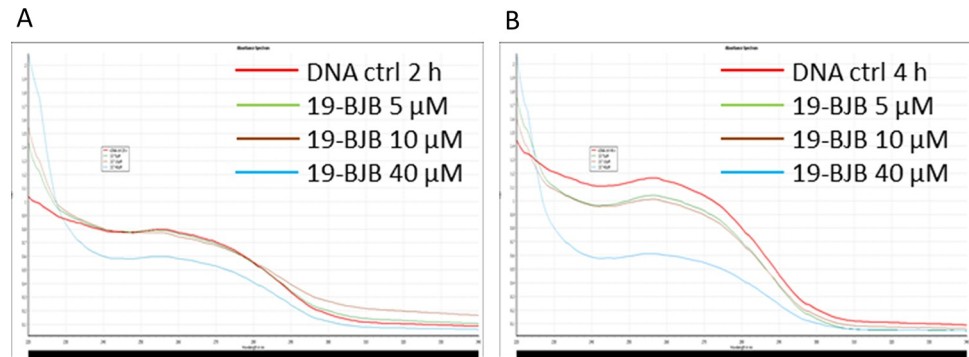

**Fig 10. The results of live and dead test of cells affected by 19-BJB and ATMI, detected by TailTM Image-Based Cytometer.** Green column represented the number of live cells and red column represented the number of dead cells. Photos of T24 live cells were stained green by 4 μM calcein AM and dead cells were stained red by 2 μM ethidium homodimer. The raw data of fluorescence response were showed (A). Cells were treated by two ways, one for treated with 19-BJB for 48 h (0, 4, 8 μM), the other treated with 10 μM ATMI first, then 19-BJB for 48 h (4, 8 μM). The histogram data (C) are expressed as means ± SD from triplicate samples for each treatment. ****p < 0.0001.

## The molecular docking study of 19-BJB and DNA

The maximum absorption wavelength of DNA is 260 nm in the ultraviolet region [40]. The presence of 19-BJB (Fig 11) could significantly decrease the UV absorption upon DNA

**Fig 11. UV-visible (220 nm—350 nm) absorption spectra of the 19-BJB conjugated with DNA at different concentrations.** (A) DNA fragment was treated with 19-BJB at different concentration (0, 5, 10, 40 μM) for 2 h. (B) DNA fragment was treated with 19-BJB at different concentration (0, 5, 10, 40 μM) for 4 h.

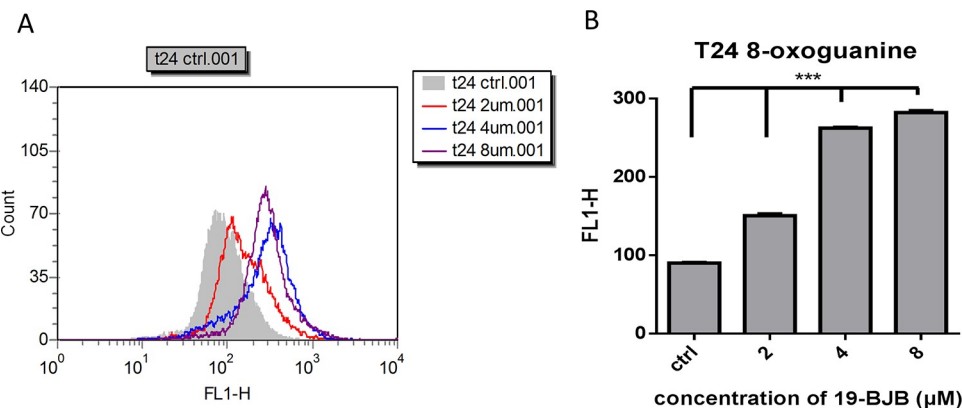

**Fig 12. Fluorescence response of 8-oxoguanine in T24 cells after treated with 19-BJB for 24 h.** (A) The FL1-H average values of deferent groups detected by flow cytometry. (B) The results were performed in column bar graphs. Error bars represent standard error of the mean, *** $p < 0.001$.

binding, and the reducing effect varies with concentration of 19-BJB. The decrease in DNA absorption was obvious after treated with 19-BJB for 4 h, especially at the concentration of 40 μM (the blue line).

The expression of DNA damage related proteins proved the possible reason of cell death caused by 19-BJB. In Fig 11, 19-BJB was observed to have an interaction with DNA, but the reason that elicited a DNA damage response was still unknown. 8-oxoguanine is one of the most common DNA lesions, which can results in a mismatched pairing with adenine resulting in G to T and C to A substitutions in the genome [41, 42]. In Fig 12, the gray shaded part represents the control group. The peak was shifted to the right as the concentration of 19-BJB was increased, indicating that the FL1-H fluorescence expression of 8-oxoguanine became more significant. The binding between 19-BJB and DNA caused the synthesis of 8-oxoguanine in T24 cells after 24 h. The results in T24 cells showed an obvious change over 4 μM of 19-BJB.

The molecular docking simulation was performed to further understand the binding situation between 19-BJB and DNA structure (CGATCG) by AutoDock software. The results showed that the value of binding energy was -5.93 kcal/mol, and the value of inhibition constant (Ki) was 44.66 μM. Based on the position at which the 19-BJB molecule docked with the DNA fragment, it was clear that 19-BJB was inserted into the CG fragment of the DNA sequence. As shown in Fig 13, the 19-BJB molecule was aligned parallel to the nucleotide chain. Besides, the carbonyl residue on the ring D (16-carbon) formed a hydrogen bond with the nucleotide chain. Furthermore, it was observed that the benzyl ester group at 19-C was inserted into the CG fragment of the DNA, and the benzene ring remained parallel to the purine ring and the pyrimidine ring of the DNA, forming a strong π-π bond, which played an important role in the docking with the DNA. In addition, the adjacent carbonyl residue (19-carbon) of the benzyl ester group also formed a hydrogen bond with the nucleotides, further stabilizing the docking effect.

### *In vivo* antitumor efficacy study

Since 19-BJB possessed a strong anti-proliferation effect, we proceeded to evaluate its antitumor efficacy *in vivo*. A T24 xenograft model [43, 44] was used because this cell line was more sensitive to 19-BJB. Preliminary studies indicated that mice were able to tolerate the acute (single-dose) administration of 19-BJB at doses of 40 mg/kg body weight. Because treatment of tumor-bearing animals was anticipated to require repeated administration, we conducted a

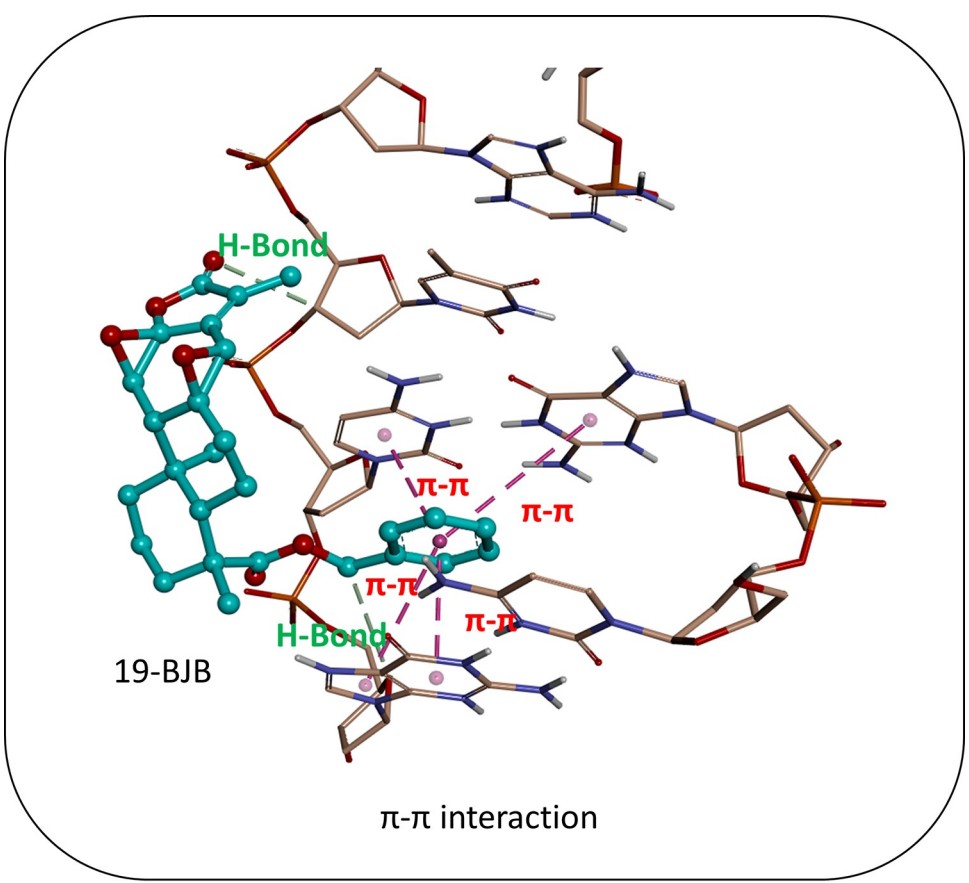

DNA Sequence: CGATCG          PDB ID: 1Z3F

**Fig 13. The molecular docking study of 19-BJB and DNA.** The figure showed the simulation that 19-BJB bound into the DNA structure (CGATCG) and formed two hydrogen bond interactions and four π-π interactions with each other. The structure of DNA was shown represented in sticks model with atom-based color (brown for C, red for O, blue for N, white for H, and dark brown for P), whereas 19-BJB was represented in sticks model with atom-based color (light blue for C and red for O).

22-day subacute toxicity study at this dose every two days (S2 Fig). The efficacy of 19-BJB in the model is summarized in Fig 14. The weights of tested mice were showed in S3 Fig. 19-BJB was well tolerated and no body-weight loss was observed at a dosage of 20 mg/kg. Compared with the vehicle treatment, 19-BJB induced inhibition (tumor growth inhibition rate = 51.7% at 20 mg/kg). The immunohistochemistry results for p-Chk1, p-Chk2, cleaved caspase-3, cleaved PARP-1, TUNEL and Ki-67 are shown in Fig 14C, and the results for the drug group obviously differed from the results for the control group.

## 3D-QSAR study of jolkinolide derivatives

After jolkinolide derivatives synthesized, comparative molecular field analysis (CoMFA) was employed to establish the 3D-QSAR model to further explain the structure-activity relationships (SAR) of jolkinolide derivatives. For the evaluation of functional groups introduced into the ring A of jolkinolide, we evaluated the 33 derivatives we had synthesized to analyze the impact on the design of jolkinolide derivatives [22, 23]. The structures of 33 jolkinolide derivatives were displayed in the S1 Fig. Ring A and ring B, the common core structure of jolkinolide

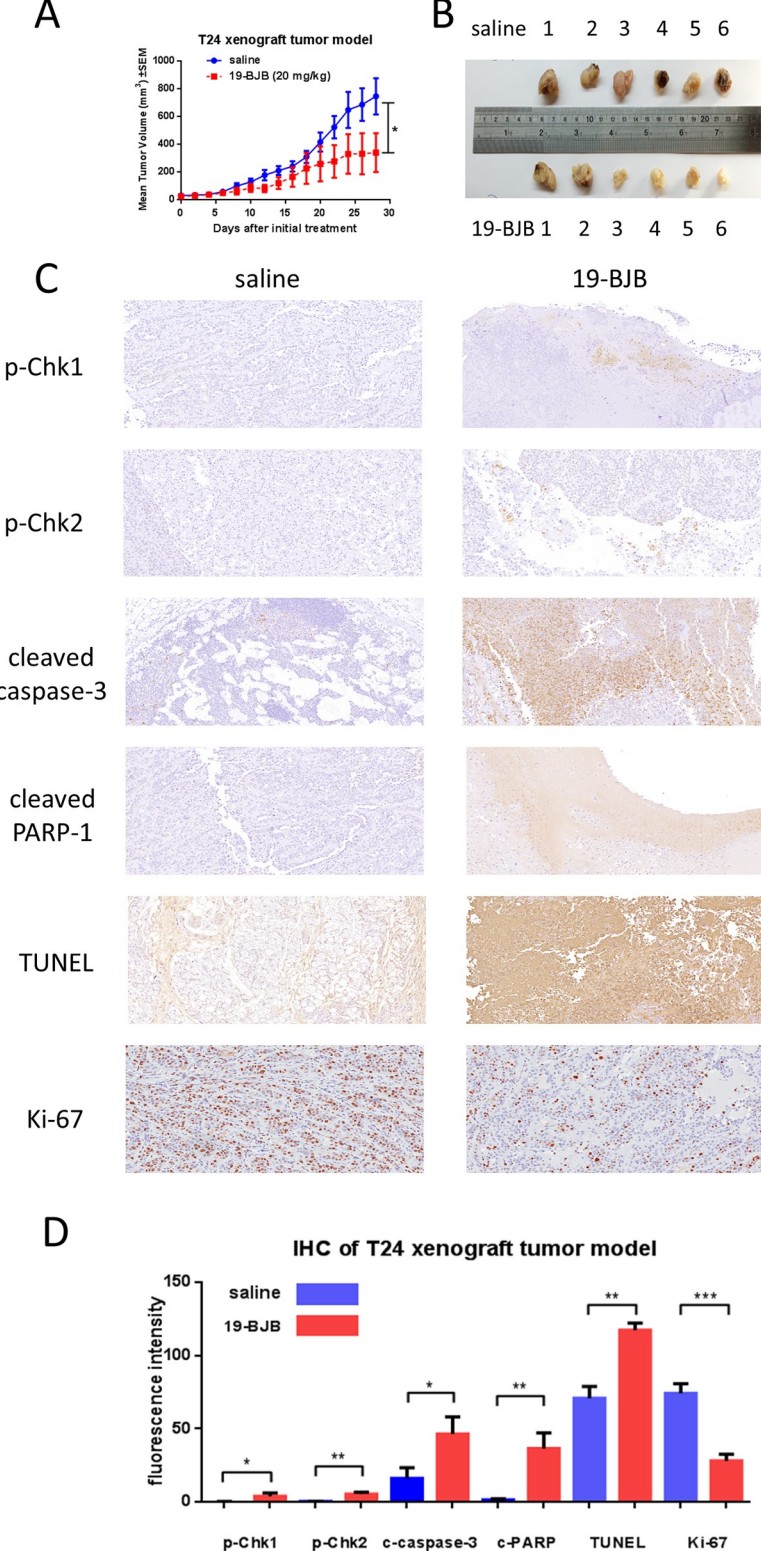

**Fig 14. The xenograft tumor model test in nude mice.** (A) Tumor volume (mm3) in nude BALB/c mice following s.
c. injection of T24 cells, control group and drug group. (B) Tumor photos (control and drug group). (C) Pictures of
immunohistochemistry for p-Chk1, p-Chk2, cleaved caspase-3, cleaved PARP-1, TUNEL, Ki-67 respectively. (D) The
IHC data was calculated by CaseViewer 2.3. Values represent the mean ± S.E.M. for 3 animals (* p < 0.05, ** p < 0.01,
mean ± S.E.M.).

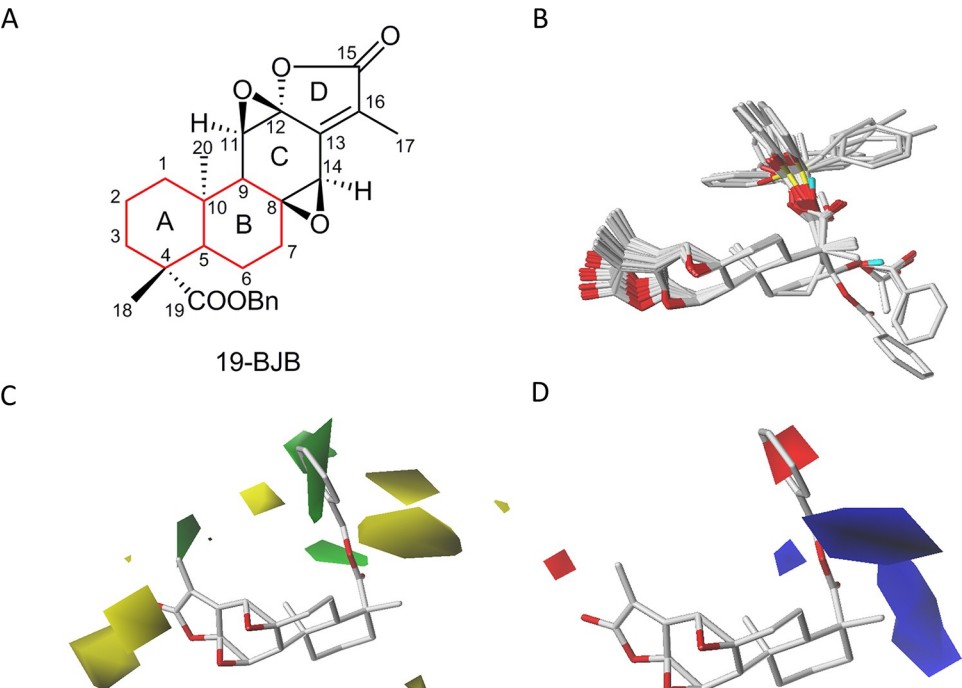

**Fig 15. CoMFA study of jolkinolides.** (A) The structure of the 19-BJB. The common substructure used for alignment was marked as red part. (B) Superposition of the 33 derivatives of jolkinolides onto the alignment hypothesis. Contour maps of the CoMFA were established using PLS analysis. 19-BJB (stick model) was chosen as a reference to depict the field region. (C) Contour map of the steric field. Green areas showed the favored steric interactions from the ligands, and the yellow areas showed the disfavored steric contributions. (D) Contour maps of the electrostatic field. Blue areas presented the favored electrostatic regions. Red areas showed the disfavored electrostatic regions.

derivatives, were chosen to set as the core structure for the alignment of 33 derivatives (Fig 15A). After that, the alignment of 33 derivatives was shown in Fig 15B. Besides, the steric field and electrostatic field of CoMFA models were respectively established and shown in Fig 15C and 15D. Additionally, the result of cross-validation statistical parameters of CoMFA model in partial least squares (PLS) analysis was summarized in the S1 Table. The predicted $pIC_{50}$ of the jolkinolide derivatives were shown in S2 Table. All the residuals between predicted and actual $pIC_{50}$ were less than a logarithmic unit, which indicated the CoMFA model had a good predictive performance. The correlation plot of actual activity against predicted activity by the CoMFA model was illustrated in the S4 Fig. All points uniformly distributed around the regression line and R2 > 0.95, which showed a good predictive ability and accuracy of the model.

As illustrated in Fig 15C, CoMFA models suggested that the bulky groups are not suitable for presentation in the areas surrounding the ring C and ring D of jolkinolides. In contrast, a bulky substituent, such as the benzyl ester group, can be introduced onto the C-19 of jolkinolides. It is expected that a substituted group modified on ring A is a strategy for increasing the anticancer activity of jolkinolides. Most important, we also found that CoMFA contour maps presented an unfavorable region on the lactone ring (ring D) (Fig 15C). For the electrostatic contour map, the contour map showed that the 11,12-epoxide ring and the 8,14-epoxide ring were located in the electrostatically unfavorable region (red area), indicating that this region favors negatively charged atoms, such as an epoxide (Fig 15D). As shown in Fig 15D, we observed that the ring C of jolkinolides favors a hydrogen bond acceptor, indicated by the magenta area located around both epoxide rings. This finding supports the electrostatic results

that the existence of an epoxide ring may contribute to the anticancer activity of the jolkino-lides' skeleton. Additionally, the substituent at C-19 is an unfavorable hydrogen bond acceptor (Fig 15D). Taking together, we propose that the substituent at C-19 is suitable for modification with a benzyl group.

## Discussion

The five-years overall survival rate of patients who suffer from bladder cancer disease has remained in the range of 60–70% over the last two decades, even the patients receive the standard of care high dose chemotherapy combined with surgical resection [5]. Although the maximal dose of conventional chemotherapy drugs has been used, the clinical outcomes of bladder cancer have not improved significantly. Some drugs, such as cisplatin and paclitaxel, are lack of specificity so that they have to be used in an excessive dose and sometimes caused drug-resistant problem [45]. In this study, we have first shown that 19-BJB exhibits the high potential of inhibitory activity against bladder cancer cells, including cisplatin- and paclitaxel-resistant cells. The *in vivo* antitumor efficacy study also indicated that 19-BJB had a good activity in inhibiting tumor growth, which encouraged that 19-BJB could be used as a potential candidate for bladder cancer therapy. In addition, the inhibitory effects of 19-BJB against cisplatin-resistant bladder cancer NP14 and paclitaxel-resistant bladder cancer NTaxol were much better than the effect of 19-BJB against its parental bladder cancer NTUB1. These findings may provide an important clue for further clinical drugs combination approaches.

For the investigation of mode of actions, we evaluated the effects of DNA damage response (DDR) upon 19-BJB treatment in the bladder cancer cells. DDR recognizes DNA damage, and then initiates a cascade reaction to repair the broken chain. ATM plays an important role in DNA damage, especially after DSBs (double-strand breaks). ATM is recruited to the sites of DSBs by sensor protein. Then the signals are transduced to downstream effectors such as Chk2 and p53, leading to cell cycle arrest [46]. ATR (ATM and Rad3-related) is another checkpoint related kinase which is related to ATM activated in response to persistent single-stranded DNA and involve in various kinds of DNA damage, especially those related to DNA replication [47]. Once ATR is activated, Chk1 is phosphorylated, initiating a signal transduction cascade that culminates in cell cycle arrest [48]. Chk1 and Chk2 coordinate the DNA damage response and cell cycle checkpoint response [49]. In early period of DSBs, H2AX is involved in the steps recruiting a series of DDR related proteins to the injury site of DNA strands after phosphorylated by ATM and ATR [50, 51]. Then p-H2AX binds to MDC1 (mediator of DNA damage checkpoint protein 1) to form a complex for the further interactions in DBS repair [52]. The presence of p-H2AX is the direct evidence of DBS in DNA [53]. PARP-1 participates in several DNA repair processes, especially plays an important role in the repair of single-stranded DNA breaks [54].

In our research, the results indicated that 19-BJB treatment could inhibit the proliferation of human bladder cancer cells via induction of DNA damage. In Figs 7 and 9, the expression of p-Chk1 and p-Chk2 increased, which indicated that cell cycle arrest. p-H2AX is also increased both in 24 h and 48 h groups, which showed a DSBs may occur. The presence of cleaved-PARP-1 proved that DNA repair was activated after the damage caused by 19-BJB. In order to clarify the mechanism of DNA damage caused by 19-BJB, KU-55933, an ATM inhibitor (ATMI), was added into medium before the cells was treated with 19-BJB. In Fig 8, the increasing expression of p-H2AX, p-Chk1 and p-Chk2 could be reversed by ATMI. These findings suggested that 19-BJB does have DNA damage and may activate ATM-related pathway. The live and dead test in Fig 10 also proved that cell viability decreased slightly when ATMI was added. The FACS analysis of cell cycle (Fig 5) showed obvious G1/S arrest in 24 h group, and

not obvious G2/M arrest in 48 h group. Considering the differences above, we tried to make a reasonable assumption. The expression of p-H2AX and cleaved-PARP-1 were checked after treating with 19-BJB in 24 h (Fig 9). At the concentration of 8 μM, an obvious increasing expression of protein was found. In Fig 12, the appearance of 8-oxoguanine in T24 cells after treating with 19-BJB in 24 h indicated that 19-BJB may cause the change in the basic structure of DNA. In early period after 19-BJB was treated, ATM/Chk2 checkpoint pathway and phosphorylation of H2AX were both activated by DSBs. Then the downstream protein cdc25A was activated, causing G1/S cell cycle arrest. Over time, ATR/Chk1 checkpoint pathway was activated by ATM, the downstream protein cdc25C was activated, causing G2/M cell cycle arrest [55, 56]. In our western blot analysis, the protein expression of p-Chk1 was significantly less than that of p-Chk2 (Fig 7A), which may indicate ATM/Chk2/cdc25A checkpoint pathway was more important in our results. The apoptosis peak became more obvious as the concentration of 19-BJB was increased, especially in the cells after 48 h treated. Additionally, the observation of apoptosis after 19-BJB treatment was monitored by the TUNEL-positive staining, and the cleaved caspase-3 (Figs 7B and 14C). In summary, we considered that the T24 cell cycle arrest caused by DNA damage would lead to cell apoptosis after 24–48 h.

19-BJB had a better inhibitory activity compared with the parent compound jolkinolide B, which may due to the modification of ring A. In previous reports, jolkinolide B was found to inhibit the growth of multiple cancer cells through different mechanisms [15–17]. Both derivatives, 17-acetoxyjolkinolide B and 17-hydroxyjolkinolide B which had a substituent on C-17 of ring D, were found to have a better inhibitory activity than jolkinolide B [18–20]. But the DNA damage effects and SAR of jolkinolide derivatives were never reported. In our study, the bulky groups such as benzyl ester may make the contribution of DNA damage. Results of DNA damage responses suggest that the mechanisms may act through the interaction of 19-BJB and DNA, which also provoked canonical DNA damage responses and checkpoint activation. The UV analysis data suggested the interaction of 19-BJB and DNA (Fig 11). In summary, the mode of action of 19-BJB may insert into DNA helix structure and further induce the DNA damage responses.

We provided an important insight into the mode underlying jolkinolides derivative-mediated anticancer effects, and these findings support the hypothesis that diterpenoids may act as attractive lead structures for developing antitumor agents. To understand the SAR of related derivatives, the 3D-QSAR model was conducted by using CoMFA analysis and the result showed that bulky groups at ring A of jolkinolides. This finding was consistent with the results of the docking of 19-BJB and DNA. The benzyl ester group of 19-BJB inserted the DNA structure and induced DNA double-strand breaks. Moreover, the ring C with the epoxy group could make the structure more rigid, so that the π-π bond formed by the benzyl ester group and hydrogen bond formed by ring D would be more stable. We therefore hypothesize that the new introduced benzyl ester group at jolkinolides play a critical role on the effects of DNA damage. The new moiety of benzyl ester contributes to the interaction of jolkinolides and DNA nucleotides (Fig 13). Collectively, the steric substituents at C-19 are able to substantially enhance the anticancer activity of jolkinolides. Together with both the epoxy on ring C and the α,β-unsaturated lactone on ring D, these functional groups are prerequisites for the inhibitory effects.

Taken together, our findings first reveal the new mode of anticancer activity induced by jolkinolides structure. The results prove that structural modification of jolkinolides could be focused on C-19 of ring A. The new modified ring A may enhance the DNA binding activity and contribute to the anticancer effects of jolkinolides.

## Supporting information

**S1 Checklist. The ARRIVE guidelines checklist.**
(PDF)

**S1 Fig. Chemical structures of 33 jolkinolide derivatives.**
(DOCX)

**S2 Fig. Toxicity test of 19-BJB in nude mice.**
(DOCX)

**S3 Fig. Mice weight after initial treatment.**
(DOCX)

**S4 Fig. Plot of actual activity against predicted activity by the CoMFA model.**
(DOCX)

**S5 Fig. The chemical structure of ATM inhibitor (KU-55933).**
(DOCX)

**S1 Table. Statistical parameters of CoMFA model.**
(DOCX)

**S2 Table. The actual and predicted pIC50 values of the jolkinolide derivatives.**
(DOCX)

**S1 Raw images.**
(PDF)

**S1 File. Affidavit of approval of animal use protocol.**
(PDF)

## Acknowledgments

We are grateful to the National Center for High-performance Computing of Taiwan for computer time and facilities.

## Author Contributions

**Conceptualization:** Juan-Cheng Yang.

**Data curation:** Ke Wang, Ya-Jing Zhang.

**Formal analysis:** Ke Wang.

**Funding acquisition:** Da-Yong Zhang, Yang-Chang Wu.

**Investigation:** Ke Wang.

**Methodology:** Juan-Cheng Yang, Yun-Hao Dai.

**Project administration:** Yang-Chang Wu.

**Visualization:** Guan-Yu Chen.

**Writing – original draft:** Ke Wang.

**Writing – review & editing:** Yeong-Jiunn Jang.

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
