## [Decision Letter · Decision Letter 0]

17 Jul 2020

PONE-D-20-08378

19-(Benzyloxy)-19-oxojolkinolide B (19-BJB), an ent-abietane diterpene diepoxide, inhibits the growth of bladder cancer T24 cells through DNA damage

PLOS ONE

Dear Dr. Wu,

Thank you for submitting your manuscript to PLOS ONE. After careful consideration, we feel that it has merit but does not fully meet PLOS ONE’s publication criteria as it currently stands. Therefore, we invite you to submit a revised version of the manuscript that addresses the points raised during the review process.

Please describe rationale and cellular phenotypes for the cell lines used in the study, particularly for the chemoresistant ones. Experiments should be performed to find out 19-BJB toxicity in non-malignant cells.  More specific inhibitors of checkpoint kinases should be used in the experiments (to avoid caffeine non-specific effects), and similar analyses (WB, fig 5) should be performed with these inhibitors (not only live-dead assay).

We look forward to receiving your revised manuscript.

Kind regards,

Irina V. Lebedeva, Ph.D.

Academic Editor

PLOS ONE

Journal Requirements:

2.PLOS ONE now requires that authors provide the original uncropped and unadjusted images underlying all blot or gel results reported in a submission’s figures or Supporting Information files. This policy and the journal’s other requirements for blot/gel reporting and figure preparation are described in detail at https://journals.plos.org/plosone/s/figures#loc-blot-and-gel-reporting-requirements and https://journals.plos.org/plosone/s/figures#loc-preparing-figures-from-image-files. When you submit your revised manuscript, please ensure that your figures adhere fully to these guidelines and provide the original underlying images for all blot or gel data reported in your submission. See the following link for instructions on providing the original image data: https://journals.plos.org/plosone/s/figures#loc-original-images-for-blots-and-gels.

4. Please ensure that you refer to Figure 1 in your text as, if accepted, production will need this reference to link the reader to the figure.

Reviewers' comments:

Reviewer's Responses to Questions

**Comments to the Author**

1. Is the manuscript technically sound, and do the data support the conclusions?

Reviewer #1: Yes

Reviewer #2: No

2. Has the statistical analysis been performed appropriately and rigorously? 

Reviewer #1: Yes

Reviewer #2: I Don't Know

3. Have the authors made all data underlying the findings in their manuscript fully available?

Reviewer #1: Yes

Reviewer #2: Yes

4. Is the manuscript presented in an intelligible fashion and written in standard English?

Reviewer #1: Yes

Reviewer #2: Yes

5. Review Comments to the Author

Reviewer #1: The author report in this manuscript the ancancer activity exerted by joikinolides structure. the structural modification of joikinolides is based on the c-19 of the ring a, which can enhance the dna Binding activity contributing to the anticancer effects of this class of drugs.

The manuscript is well organized, well written and easy to follow.

It would be interesting to investigate whether the treatment with this class of drugs may affect the DSBs repair by NHEJ or HR. The authors should assess the DSBs repair by utilizing specific reporter assays. The experiment performed in in vivo models support the in vitro investigations.

In the discussion section (lines 488-491) the manuscript could take advantage by the mention to novel combinatorial strategies recently introduced or proposed for urothelial cancer treatment as recently suggested (Criscuolo et al,JECCR, 2019, doi: 10.1186/s13046-019-1089-z; Morra F et al, JECCR, 2019, doi: 10.1186/s13046-019-1087-1).

Reviewer #2: Wang et al present data on the effect of the diterpene diepoxide 19-BJB on bladder cancer cell lines as well as on its interaction with DNA. They suggest that cytotoxicity of the compound is caused by DNA damage and may be therapeutically applied. Although they demonstrate some interaction of the compound with DNA, I feel that the data provided is too indirect to permit the conclusion that 19-BJB damages DNA, least providing a mechanism of how it does, and that this damage is the primary cause of cytotoxicity. Specifically, I have the following comments:

There is no demonstration of DNA damage in vitro. As far as I understand the data, they only show binding of 19-BJB to DNA with some evidence of intercalation. However, there is no demonstration of strand-breaks or adducts. The mechanism of DNA damage needs to be clarified.

The data obtained with cell lines also does not prove the conclusion. There is a discrepancy between the concentrations required to induce DNA damage and cytotoxicity, respectively. To me, it seems that apoptosis is induced by lower concentrations than DNA damage (Fig. 5). In addition to other experiments, it would be important to establish the time course of events as well.

Likewise, the binding concentrations in vitro and in vivo do not seem to fit the bill. Have the authors considered intracellular metabolism of the compound? After all, it is an epoxide and activated metabolites could be substantially more reactive.

The analyses of cell death are incomplete. Cell cycle profiles are required. Caffeine is a quite unspecific compound and more specific inhibitors of checkpoint kinases should be used.

It should be explained, why the particular cell lines were used. Important properties of not commonly used cell lines like NT-TUB and its derivatives should be introduced (e.g. p53 status in this context) and they should be referenced. A more representative selection of bladder cancer cell lines, including more differentiated lines, would make sense. Moreover, the tumor-specificity of the treatment needs to be demonstrated, using e.g. fibroblasts and non-transformed urothelial cells.

The discussion lacks depth and among others, a comparison with findings for other compounds that may have a similar mode of action.

For the more biologically experienced reader the significance and limitations of the molecular docking and CoMFA need to be better explained; for chemists and pharmacologists their limitations need to be more critically considered.

Some editing for language is required, e.g. phosphor- should read phospho- in l.328, or l.270-272, etc.

The introductory statements on bladder cancer are not accurate; I agree that it is an important (and often underestimated) disease, but the incidence has certainly not “skyrocketed” in recent years.

6. PLOS authors have the option to publish the peer review history of their article (what does this mean?). If published, this will include your full peer review and any attached files.

Reviewer #1: No

Reviewer #2: **Yes: **Wolfgang A. Schulz

---

## [Author Response · Author response to Decision Letter 0]

10 Nov 2020

All line numbers mentioned below correspond to the document “Manuscript”.

Editor: 

1. Please describe rationale and cellular phenotypes for the cell lines used in the study, particularly for the chemoresistant ones.

Answer: The rationale and cellular phenotypes for cell lines were described in line 93-100. Most of them were searched from ATCC.

2. Experiments should be performed to find out 19-BJB toxicity in non-malignant cells.

Answer: HaCat cells were chosen to perform the toxicity of 19-BJB. The result was in Fig 2F.

3. More specific inhibitors of checkpoint kinases should be used in the experiments (to avoid caffeine non-specific effects), and similar analyses (WB, fig 5) should be performed with these inhibitors (not only live-dead assay).

Answer: We had changed all experiments about caffeine to KU-55933, an ATM inhibitor (ATMI in manuscript). This drug was purchased from Sigma-Aldrich. The chemical structure is in S5 Fig. The cell cycle test of T24 was also performed. The new experiments about cell cycle, WB and live-dead assay were shown in Fig 5, 8, and 9. The related descriptions in Materials/Methods and Results were also changed. A more detailed discussion was presented in line 541-584.

Reviewer 1: It would be interesting to investigate whether the treatment with this class of drugs may affect the DSBs repair by NHEJ or HR. The authors should assess the DSBs repair by utilizing specific reporter assays.

Answer: We had changed the caffeine (non-specific effects agent) to KU-55933, a specific ATM inhibitor to avoid confusing result. In the new experiment of live and dead (Fig 9), we found that 19-BJB had a synergistic effect with ATM inhibitor. The FACS analysis of cell cycle (Fig 5) showed obvious G1/S arrest in 24 h group, and not obvious G2/M arrest in 48 h group. The apoptosis peak became more obvious as the concentration of 19-BJB was increased, especially in the cells after 48 h treated. So we considered that the DNA damage effect of 19-BJB could not be eliminate in 48 h which means a failure of DNA repair. A more detailed discussion was presented in line 541-584.

Reviewer 2: 

1. There is no demonstration of strand-breaks or adducts. The mechanism of DNA damage needs to be clarified.

Answer: We had rewritten paragraphs about DNA damage in 541-558. Here are two reasons that make us think about DSBs (double-strand breaks) effect of 19-BJB. Firstly, ATM plays an important role in DNA damage, especially after DSBs. ATM is recruited to the sites of DSBs by sensor protein. Then the signals are transduced to downstream effectors such as Chk2/p-H2AX/p53, leading to cell cycle arrest [44]. Secondly, the presence of p-H2AX is the direct evidence of DBS in DNA [51]. In Fig 7, the expression of p-Chk2 and p-H2AX increased which showed a DSBs may occur.

2. The data obtained with cell lines also does not prove the conclusion. There is a discrepancy between the concentrations required to induce DNA damage and cytotoxicity, respectively. To me, it seems that apoptosis is induced by lower concentrations than DNA damage (Fig. 5). In addition to other experiments, it would be important to establish the time course of events as well.

Answer: To establish the time course of events is a good suggestion. In cell cycle analysis (Fig 5 in new manuscript), we had chosen 24 h and 48 h after treatment as different groups. The apoptosis peak was observed as the concentration of 19-BJB was increased in the cells after 48 h treated, but not so obvious in 24 h group. In Fig 7 of new manuscript (original Fig 5), the evidence of apoptosis is cleaved caspase-3. While cleaved PARP-1, p-Chk1, p-Chk2 and p-H2AX is related to DNA damage. In my opinion, although they have connections, cleaved caspase-3 is considered as the direct evidence. Compared to p-Chk1, p-Chk2 or cleaved PARP-1, the increasing expression of cleaved caspase-3 is not so obvious at low concentration of 19-BJB. 

3. Likewise, the binding concentrations in vitro and in vivo do not seem to fit the bill. Have the authors considered intracellular metabolism of the compound? After all, it is an epoxide and activated metabolites could be substantially more reactive.

Answer: It is an interesting question about the metabolites of 19-BJB. Actually, we have considered that if 19-BJB would be decomposed into 19-Acid-JB and BnOH. After all, ester always can be hydrolyzed in cells or tissues. So we had tested the effect of 19-Acid-JB (synthesized by ourselves) and BnOH (considered to have toxicity). Neither of them could inhibit cell growth. The metabolism of epoxide compound could be substantially more reactive, but we cannot synthesize. HPLC-MS may be a good way to solve this problem. The cells should be harvested after treatment with 19-BJB and detected the metabolites in MS after purified in HPLC. The further researches will be considered. Thank you for this suggestion!

4. The analyses of cell death are incomplete. Cell cycle profiles are required. Caffeine is a quite unspecific compound and more specific inhibitors of checkpoint kinases should be used.

Answer: We had changed all experiments about caffeine to KU-55933, an ATM inhibitor (ATMI in manuscript). This drug was purchased from Sigma-Aldrich. The chemical structure is in S5 Fig. The new experiments about cell cycle, WB and live-dead assay were shown in Fig 5, 8, and 9. The related descriptions in Materials/Methods and Results were also changed. A more detailed discussion was presented in line 541-584.

5. It should be explained, why the particular cell lines were used. Important properties of not commonly used cell lines like NT-TUB and its derivatives should be introduced (e.g. p53 status in this context) and they should be referenced. A more representative selection of bladder cancer cell lines, including more differentiated lines, would make sense. Moreover, the tumor-specificity of the treatment needs to be demonstrated, using e.g. fibroblasts and non-transformed urothelial cells.

Answer: The rationale and cellular phenotypes for cell lines were described in line 93-100. Most of them were searched from ATCC. NTUB1, NP14 and NTaxol were chosen because these kinds of bladder cell lines have the most stable drug-resistant effects in our lab. When we finished the synthesis of jolkinolide derivatives to obtain more than 30 kinds of compounds (S1 Fig), it is very important to test the potential of jolkinolide structure about combination medication for the further researches. Cisplatin is commonly used drug in clinical treatment of bladder cancer. Taxol, a natural product same as jolkinolide A and B, also have a good effect in cancer therapy. T24 and J82 are two widely used cell lines which are often used in animal model.

6. The discussion lacks depth and among others, a comparison with findings for other compounds that may have a similar mode of action.

Answer: We had rewritten the paragraph of discussion. The results and detailed discussions of new experiments were presented in line 541-584.

7. For the more biologically experienced reader the significance and limitations of the molecular docking and CoMFA need to be better explained; for chemists and pharmacologists their limitations need to be more critically considered.

Answer: We had rewritten some sentences to make it easier to understand. In this article, QSAR is a highlight that is the first time to explain the detail structure-activity relationship of jolkinolide derivatives, so we kept and revised this part.

8. Some editing for language is required, e.g. phosphor- should read phospho- in l.328, or l.270-272, etc. The introductory statements on bladder cancer are not accurate; I agree that it is an important (and often underestimated) disease, but the incidence has certainly not “skyrocketed” in recent years.

Answer: We had revised the expression mention above and made the manuscript more accurate.

---

## [Decision Letter · Decision Letter 1]

18 Dec 2020

PONE-D-20-08378R1

19-(Benzyloxy)-19-oxojolkinolide B (19-BJB), an ent-abietane diterpene diepoxide, inhibits the growth of bladder cancer T24 cells through DNA damage

PLOS ONE

Dear Dr. Wu,

Thank you for submitting your manuscript to PLOS ONE. After careful consideration, we feel that it has merit but does not fully meet PLOS ONE’s publication criteria as it currently stands. Therefore, we invite you to submit a revised version of the manuscript that addresses the points raised during the review process.

Please address the reviewer comments promptly.

Provide better quality figures.

We look forward to receiving your revised manuscript.

Kind regards,

Irina V. Lebedeva, Ph.D.

Academic Editor

PLOS ONE

Reviewers' comments:

Reviewer's Responses to Questions

**Comments to the Author**

1. If the authors have adequately addressed your comments raised in a previous round of review and you feel that this manuscript is now acceptable for publication, you may indicate that here to bypass the “Comments to the Author” section, enter your conflict of interest statement in the “Confidential to Editor” section, and submit your "Accept" recommendation.

Reviewer #1: (No Response)

Reviewer #2: (No Response)

2. Is the manuscript technically sound, and do the data support the conclusions?

Reviewer #1: Partly

Reviewer #2: Partly

3. Has the statistical analysis been performed appropriately and rigorously? 

Reviewer #1: N/A

Reviewer #2: Yes

4. Have the authors made all data underlying the findings in their manuscript fully available?

Reviewer #1: Yes

Reviewer #2: Yes

5. Is the manuscript presented in an intelligible fashion and written in standard English?

Reviewer #1: Yes

Reviewer #2: No

6. Review Comments to the Author

Reviewer #1: Despite the authors have not completely addressed this reviewer requests, the revised manuniscript is certainly improved, and is now suitable for publication.

Mention to the suggested reccomended papers would have been appreciated.

Reviewer #2: The manuscript is much improved; the data are better explained and the additional experiments have helped to clarify the matter. Several issues remain, however.

Major comments:

I feel that the authors have still not excluded the alternative interpretation of their findings that DNA damage is only part of the mode of action of 19BJB (and perhaps only indirectly so). They observe an increased G1 fraction after 24 h which is followed by apoptosis later. All data on DNA damage response were however obtained at 48 h. At least increased H2A.X phosphorylation should be seen at 24 h. Positive comet assays can also result from (early stages of) apoptosis. 19BJB could therefore act via another mechanism.

This alternative is the more likely as the data (little of which is truly experimental) show binding to DNA and insertion. Thus, the structure of DNA may be disturbed, but does this suffice to elicit a DNA damage response? This should be shown or at least discussed by comparison with other compounds binding to DNA in a similar manner. For instance, bleomycin intercalates (even more deeply) into DNA, but DNA damage is actually caused by a part of the molecule generating hydroxyl radicals. On a related point: how do the authors interpret the change in the UV spectrum? Can this change be explained by insertion and disturbance of pi-pi interactions or would it require unwinding?

Minor comments:

Another round of language editing is required, especially in the abstract, but also throughout the manuscript.

Many figures in the pdf are not of sufficient quality. Much better quality needs to be provided in the final version.

Throughout the manuscript, including the figures, numbers and units should be separated by a space.

Throughout the manuscript, round off numbers to a reasonable number of digits. In particular, table 1 would be better legible if corrected in this way.

The first § on bladder cancer is still inaccurate and the references are not recent or do not fit well.

Many other statements in the manuscript are also inaccurate, in many cases because the wording is incorrect. Proper scientific terms are especially important. Thus, in cell cycle analysis, “rates” should be replaced by “fractions” (the entire § from l.341 needs correction); chromatin is not “depolymerized” (l.557), “conjugation” is not correct for binding to and insertion into DNA. Please check throughout.

l.573f: synergy has not been shown.

7. PLOS authors have the option to publish the peer review history of their article (what does this mean?). If published, this will include your full peer review and any attached files.

Reviewer #1: No

Reviewer #2: **Yes: **Wolfgang A. Schulz

---

## [Author Response · Author response to Decision Letter 1]

5 Feb 2021

All line numbers mentioned below correspond to the document “Manuscript”.

Reviewer 1: Despite the authors have not completely addressed this reviewer requests, the revised manuscript is certainly improved, and is now suitable for publication. Mention to the suggested recommended papers would have been appreciated.

Answer: Thank you for your suggestions before. We had revised the manuscript according to the comments.

Reviewer 2: The manuscript is much improved; the data are better explained and the additional experiments have helped to clarify the matter. Several issues remain, however. Major comments:

1. I feel that the authors have still not excluded the alternative interpretation of their findings that DNA damage is only part of the mode of action of 19BJB (and perhaps only indirectly so). They observe an increased G1 fraction after 24 h which is followed by apoptosis later. All data on DNA damage response were however obtained at 48 h. At least increased H2A.X phosphorylation should be seen at 24 h. Positive comet assays can also result from (early stages of) apoptosis. 19BJB could therefore act via another mechanism.

Answer: We did not exclude the alternative interpretation, because of the observation of increasing G1 fraction after 24 h. So the expressions of phosphorylation of H2A.X and PARP-1 were checked by western blot after treatment with 19-BJB for 24 h (Line 423-433). The expression of cleaved-PARP-1 was increased in 4 μM and 8 μM groups. The expression of p-H2A.X was also significantly increased in 8 μM group. Taken together, we think that the increasing G1 fraction after 24 h is also caused by DNA damage.

2. This alternative is the more likely as the data (little of which is truly experimental) show binding to DNA and insertion. Thus, the structure of DNA may be disturbed, but does this suffice to elicit a DNA damage response? This should be shown or at least discussed by comparison with other compounds binding to DNA in a similar manner. For instance, bleomycin intercalates (even more deeply) into DNA, but DNA damage is actually caused by a part of the molecule generating hydroxyl radicals. On a related point: how do the authors interpret the change in the UV spectrum? Can this change be explained by insertion and disturbance of pi-pi interactions or would it require unwinding?

Answer: Thank you for your suggestion. At the beginning, we did think that the changes of DNA structure may lead to DNA damage. But bleomycin is a good example to prove that our previous concept was inaccurate. The changes in the UV spectrum are more like “interaction” or “intercalates” but not the evidence of DNA damage. 

In this revised manuscript, we checked the expression of 8-oxoguanine (Line 463-477), for it is one of the most common DNA lesions, which can results in a mismatched pairing with adenine resulting in G to T and C to A substitutions in the genome. As the concentration of 19-BJB was increased, the expression of 8-oxoguanine became more significant. This finding indicated that the interaction between 19-BJB and DNA may cause the synthesis of 8-oxoguanine in T24 cells after 24 h. The structural changes of DNA may be the main reason of damage. 

Minor comments:

3. Another round of language editing is required, especially in the abstract, but also throughout the manuscript.

Answer: We have revised the words in the manuscript and tried to make it be accurate, especially in abstract.

4. Many figures in the pdf are not of sufficient quality. Much better quality needs to be provided in the final version.

Answer: The figures were reorganized to be quality.

5. Throughout the manuscript, including the figures, numbers and units should be separated by a space.

Answer: We have rechecked the manuscript and revised the missing parts again.

6. Throughout the manuscript, round off numbers to a reasonable number of digits. In particular, table 1 would be better legible if corrected in this way.

Answer: We have round off numbers to a reasonable number of digits in table 1 (Line 302-305), but skipped the data of paclitaxel in T24 and NP14 cell lines as the numbers are too small. The other parts of manuscript were also rechecked.

7. The first § on bladder cancer is still inaccurate and the references are not recent or do not fit well.

Answer: We have replaced the references with recent articles and rewritten paragraph 1 in introduction (Line 41-44). 

8. Many other statements in the manuscript are also inaccurate, in many cases because the wording is incorrect. Proper scientific terms are especially important. Thus, in cell cycle analysis, “rates” should be replaced by “fractions” (the entire § from l.341 needs correction); chromatin is not “depolymerized” (l.557), “conjugation” is not correct for binding to and insertion into DNA. Please check throughout. l.573f: synergy has not been shown.

Answer: We have changed the words to the accurate expression. For example, “rates” was replaced by “fractions”, “conjugation” was replaced by “interaction”. The sentence related “depolymerized” was modified and “synergy” was deleted, respectively.

---

## [Decision Letter · Decision Letter 2]

1 Mar 2021

19-(Benzyloxy)-19-oxojolkinolide B (19-BJB), an ent-abietane diterpene diepoxide, inhibits the growth of bladder cancer T24 cells through DNA damage

PONE-D-20-08378R2

Dear Dr. Wu,

We’re pleased to inform you that your manuscript has been judged scientifically suitable for publication and will be formally accepted for publication once it meets all outstanding technical requirements.

Kind regards,

Irina V. Lebedeva, Ph.D.

Academic Editor

PLOS ONE

Additional Editor Comments (optional):

Reviewers' comments:

Reviewer's Responses to Questions

**Comments to the Author**

1. If the authors have adequately addressed your comments raised in a previous round of review and you feel that this manuscript is now acceptable for publication, you may indicate that here to bypass the “Comments to the Author” section, enter your conflict of interest statement in the “Confidential to Editor” section, and submit your "Accept" recommendation.

Reviewer #1: All comments have been addressed

Reviewer #2: (No Response)

2. Is the manuscript technically sound, and do the data support the conclusions?

Reviewer #1: (No Response)

Reviewer #2: Yes

3. Has the statistical analysis been performed appropriately and rigorously? 

Reviewer #1: (No Response)

Reviewer #2: Yes

4. Have the authors made all data underlying the findings in their manuscript fully available?

Reviewer #1: (No Response)

Reviewer #2: Yes

5. Is the manuscript presented in an intelligible fashion and written in standard English?

Reviewer #1: (No Response)

Reviewer #2: Yes

6. Review Comments to the Author

Reviewer #1: (No Response)

Reviewer #2: Please check the manuscript once more thoroughly for clarity and language to optimally convey your findings to the readers.

7. PLOS authors have the option to publish the peer review history of their article (what does this mean?). If published, this will include your full peer review and any attached files.

Reviewer #1: No

Reviewer #2: **Yes: **Wolfgang A. Schulz

---

## [Editor Report · Acceptance letter]

4 Mar 2021

PONE-D-20-08378R2 

19-(Benzyloxy)-19-oxojolkinolide B (19-BJB), an *ent*-abietane diterpene diepoxide, inhibits the growth of bladder cancer T24 cells through DNA damage 

Dear Dr. Wu:

I'm pleased to inform you that your manuscript has been deemed suitable for publication in PLOS ONE. Congratulations! Your manuscript is now with our production department. 

Kind regards, 

on behalf of

Dr. Irina V. Lebedeva 

Academic Editor

PLOS ONE